# Patterns of genetic differentiation and the footprints of historical migrations in the Iberian Peninsula

Clare Bycroft [1], Ceres Fernandez-Rozadilla[2], Clara Ruiz-Ponte[2], Inés Quintela[3], Ángel Carracedo[2,3,4], Peter Donnelly [1,5] & Simon Myers[1,5]

The Iberian Peninsula is linguistically diverse and has a complex demographic history, including a centuries-long period of Muslim rule. Here, we study the fine-scale genetic structure of its population, and the genetic impacts of historical events, leveraging powerful, haplotype-based statistical methods to analyse 1413 individuals from across Spain. We detect extensive fine-scale population structure at extremely fine scales (below 10 Km) in some regions, including Galicia. We identify a major east-west axis of genetic differentiation, and evidence of historical north to south population movement. We find regionally varying fractions of north-west African ancestry (0–11%) in modern-day Iberians, related to an admixture event involving European-like and north-west African-like source populations. We date this event to 860–1120 CE, implying greater genetic impacts in the early half of Muslim rule in Iberia. Together, our results indicate clear genetic impacts of population movements associated with both the Muslim conquest and the subsequent *Reconquista*.

[1] Wellcome Trust Centre for Human Genetics, University of Oxford, Oxford OX3 7BN, UK. [2] Fundación Pública Galega de Medicina Xenómica- CIBERER-IDIS, Santiago de Compostela, A Coruña, Spain. [3] Grupo de Medicina Xenómica, Centro Nacional de Genotipado (CEGEN-PRB2-ISCIII), Universidade de Santiago de Compostela, A Coruña, Spain. [4] Institute of Forensic Sciences., University of Santiago de Compostela, A Coruña, Spain. [5] Department of Statistics, University of Oxford, Oxford OX1 3LB, UK. These authors jointly supervised this work: Peter Donnelly, Simon Myers. Correspondence and requests for materials should be addressed to S.M. (email: myers@stats.ox.ac.uk)

Genetic differentiation within or between human populations (population structure) has been studied using a variety of approaches over many years[1–5]. Recently there has been an increasing focus on studying genetic differentiation at fine geographic scales, such as within countries[6–8]. Identifying such structure allows the study of recent population history, and identifies the potential for confounding in association studies, particularly when testing rare, often recently arisen variants[9]. The Iberian Peninsula is linguistically diverse, has a complex demographic history, and is unusual among European regions in having a centuries-long period of Muslim rule[10].

Previous studies of population structure in Spain have examined either a small fraction of the genome[11–13] or only a few regions of Spain[14,15], and typically compare groups of individuals defined a priori using broad ethnic or geographic labels, such as autonomous community. Using such approaches only limited population structure within Iberia has been identified[15–19]. Some structure within northern Spain has been detected, including statistically significant differences in frequencies of Y-chromosome haplotypes and other genetic markers between the Basque-speaking regions and other parts of Iberia[11,12], a result consistent with a European-wide analysis using autosomal DNA[20]. Studies of Spain that used genome-wide data did not leverage information in correlations between genetic markers[14,15], excepting one study[21], which detected a cline of variation broadly distinguishing samples in País Vasco from other parts of northern Spain, especially Galicia, but no evidence of sub-structure in central or southern Spain. Thus the overall pattern of population structure within Spain—including subtle structure at fine geographic scales—remains uncharacterized.

The cultural and linguistic impact of Muslim rule in Iberia is well-documented, but the historical record is limited in its ability to inform about the extent, timing and geographic spread of genetic mixing between immigrants and indigenous Iberians over several centuries after the initial conquest[22]. Previous genetic studies have reported signals of admixture from sub-Saharan Africa and/or north Africa into Iberia at some point in the past[23–27]. However, estimates of the timing of this admixture vary greatly, from as long as 74 generations ago (~100 BC)[23] to 23 generations ago (~1330 CE)[25]. Estimates of overall mean proportions of African-like DNA in the Iberian Peninsula also vary, ranging from 2.4[24] to 10.6%[11]. Differences within Iberia have also been reported[11,26], based on comparisons between sampled regions, with higher fractions observed in western regions of Iberia (e.g. 21.7% in Northwest Castile[11]) and lower fractions in the north-east (e.g. 2.3% in Cataluña[11]). Estimates of the timing and extent of admixture tend to vary depending on the reference populations assumed to represent the ancestral mixing groups (e.g. Moroccan[11] or Saharawi[26]), as well as heterogeneity in the ancestral make-up of the modern-day Iberian samples used in the analysis.

Here we analyse genome-wide genotyping array data for 1413 Spanish individuals sampled from across Spain. By using powerful, haplotype-based statistical methods we identify extensive fine-scale structure down to scales <10 km in some places. We identify a major axis of genetic differentiation that runs from east to west across Iberia. In contrast, we observe remarkable genetic similarity in the north–south direction, and evidence of historical north–south population movement. Finally, we sought to clarify the timing and composition of African-like and potentially non-African genetic contributions to the Iberian Peninsula, by jointly analysing genotype data sourced from a wide range of African and European regions. We show that modern Spanish people have regionally varying fractions of ancestry from a group most similar to modern north-west Africans. This African ancestry, identified without making particular prior assumptions about

source populations, results from an admixture event that we date to 860–1120 CE, corresponding to the early half of Muslim rule. Our results indicate that it is possible to discern clear genetic impacts of the Muslim conquest and population movements associated with the subsequent *Reconquista*.

## Results

**Extensive fine-scale population structure in Spain.** We analysed phased genotyping array data for 1413 Spanish individuals typed at 693,092 autosomal single nucleotide polymorphisms (SNPs) after quality control (Methods). We applied fineSTRUCTURE[28] to these data to infer clusters of individuals with similar patterns of shared ancestry (Methods). fineSTRUCTURE inferred 145 distinct clusters, along with a hierarchical tree describing relationships between the clusters (Fig. 1a; Methods). We used genetic data only in the inference, but explored the relationship between genetic structure and geography using a subset of 726 individuals for whom geographic information was available and all four grandparents were born within 80 km of the centroid of their birthplaces. Figure 1b represents each of these individuals as a point on a map of Spain, located at the centroid of their grandparents' birthplaces and labelled according to their cluster assignment after combining small clusters at the bottom of the tree (Methods). Their grandparents were likely to have been born in the decades either side of 1900 (median birth-year of the cohort is 1941), so the spatial distribution of genetic structure described in this study would reflect that of Spain around that time.

These results reveal patterns of rich fine-scale population structure in Spain. At the coarsest level of genetic differentiation (i.e. two clusters at the top of the hierarchy) individuals located in a small region in south-west Galicia are separated from those in the rest of Spain. The next level separates individuals located primarily in the Basque regions in the north (País Vasco and Navarra) from the rest of Spain. Further down the tree (background colours in Fig. 1b) many of the clusters closely follow the east–west boundaries of Spain's autonomous communities, especially in the north of Spain. However, in the north–south direction several clusters cross boundaries of multiple autonomous communities. Overall, the major axis of genetic differentiation runs from east to west, while conversely there is remarkable genetic similarity on the north–south direction. In a complementary analysis that included Portugal, although fewer SNPs (Methods), Portuguese individuals co-clustered with individuals in Galicia (Fig. 2a), showing that this pattern extends across the whole Iberian Peninsula. Indeed, rather than mainly reflecting modern-day political boundaries (autonomous communities), the broad-scale genetic structure of the region is strikingly similar to the linguistic frontiers[29] present in the Iberian Peninsula around 1300 CE (Fig. 1c). Via more formal simulation-based testing, we confirmed this: the association of genetic structure with language is statistically significant ($p < 0.008$), even after accounting for both physical distance and autonomous community membership (Supplementary Note 9; Supplementary Figure 8). Conversely, once physical distance and language are taken into account, no significant association with autonomous community remains ($p = 0.12$).

Although some geographically dispersed clusters (e.g. 'central' and 'west') remain largely intact at the bottom of the hierarchical tree (Fig. 2b) many of the clusters that emerge further down the tree involve greater geographical localisation. By far the strongest sub-structure is seen within a single province in Galicia, Pontevedra, which contains almost half of the inferred clusters in all of Spain (Fig. 1a). This ultra-fine structure is seen across scales of <10 km and the clusters align with regions defined by

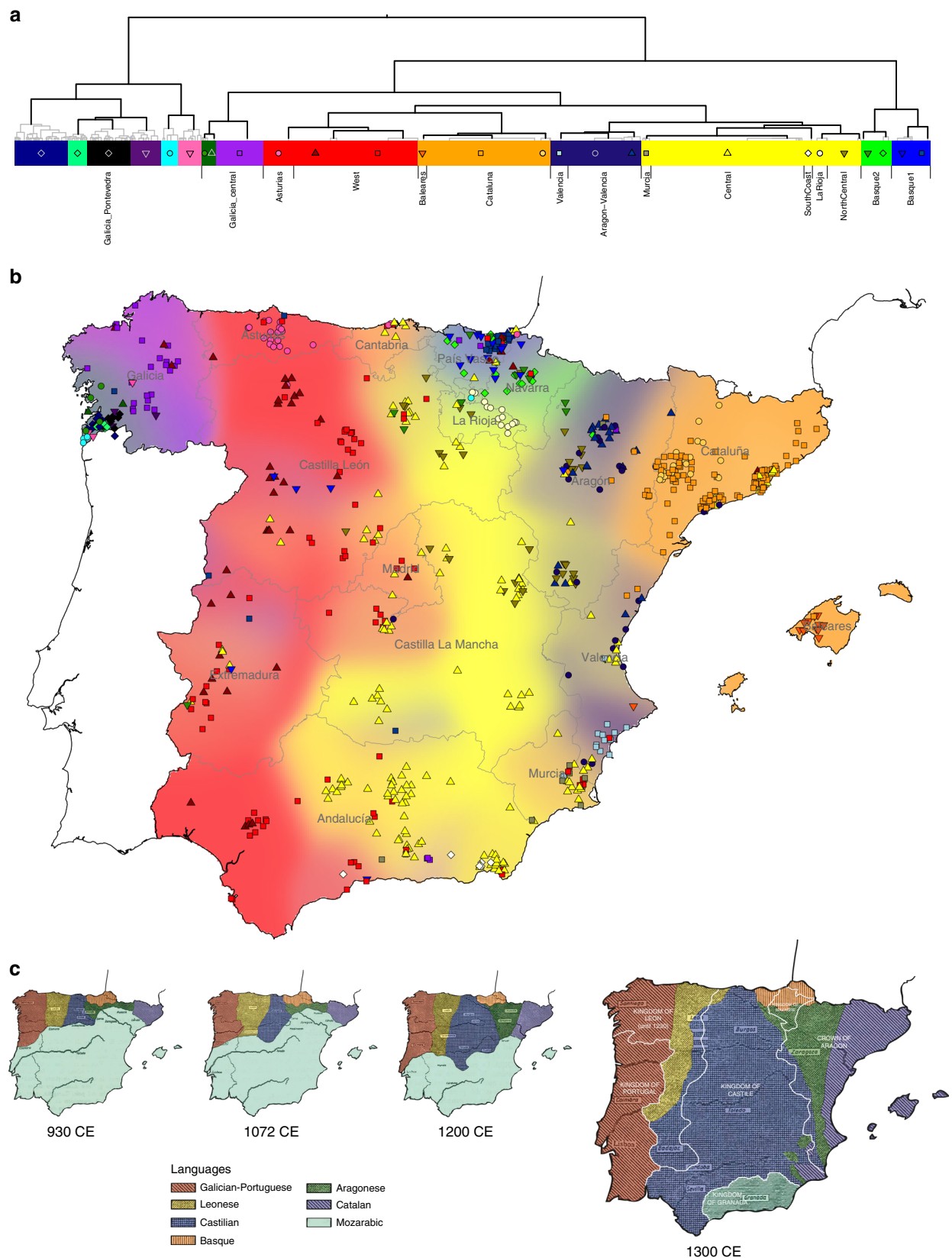

hills and/or river valleys (Fig. 3a). This structure is not an artefact of the denser sampling in this region, as it was still evident in an analysis after sub-sampling (Supplementary Note 4). Highly localised structure is also seen in other parts of Spain, including four clusters within the Basque regions (Fig. 3b), and a cluster

that is exclusive to a ~50 km segment of the River Ebro in La Rioja (Fig. 3c).

To further understand the relationships between the clusters inferred by fineSTRUCTURE, we examined patterns in the matrix of ancestry sharing (coancestry) between each pair of 1413

**Fig. 1** Spanish individuals grouped into clusters using genetic data only. **a** Binary tree showing the inferred hierarchical relationships between clusters inferred using genotype data of 1413 individuals (fineSTRUCTURE analysis A). The colours and points correspond to the clusters shown on the map, and the length of the coloured rectangles is proportional to the number of individuals assigned to that cluster. We combined some small clusters (Methods) and the thick black branches indicate the clades of the tree that we visualise in the map. Clusters are labelled according to the approximate location of most of their members, but geographic data was not used in the inference. **b** Each individual ($n = 726$) is represented by a point placed at (or close to, <24 Km) the centroid of their grandparents' birthplaces. We only plot the individuals for whom all four grandparents were born within 80 km of their average birthplace, although the data for all individuals were used in the fineSTRUCTURE inference. The background is coloured according to the spatial densities of each cluster at the level of the tree where there are 14 clusters (Methods). The colour and symbol of each point corresponds to the cluster the individual was assigned to at a lower level of the tree, as shown in **a**. Spain's autonomous communities are also shown. **c** A representation of changes in the linguistic and political boundaries in Iberia from ~930 to 1300 CE, adapted with permission from maps by Baldinger[29]. Different linguistic areas are shown with the colours and shading, and political boundaries with white borders (in the far right map only). Only the colours and labels of the Christian kingdoms have been added to aid visualisation

individuals (Fig. 4a). In general, coancestry between individuals within a cluster is higher than between individuals in different clusters, reflecting genetic drift unique to each cluster. This effect is strongest for highly localised clusters, such as those in Galicia and País Vasco and La Rioja (Fig. 4b). These clusters also tend to have greater certainty in their cluster assignment (Supplementary Figure 1b). In contrast, the cluster labelled 'central' (shown with yellow triangles in Fig. 1b) shows no clear drift signal. In fact, individuals in this cluster have—on average—more coancestry with the members of Basque-centred clusters (blue squares and triangles) than they do with other individuals in their own cluster ($p < 0.02$; Fig. 4c). Theoretical arguments predict (Methods) that this effect can only occur if admixture from a highly drifted group into another population takes place. That is, the effect could not be explained by Basques inheriting DNA from ancestors of the central group (although this may have happened in addition). Thus, this signal provides evidence of admixture into the 'central' cluster from a group related to the Basque populations.

**The genetic impact of historical migrations**. Next, we sought to characterize the relationship between Iberians (combining Spanish and Portuguese individuals) and non-Iberian groups, to understand the extent to which recent migrations from outside Iberia have influenced modern-day DNA in Spain. We constructed a combined dataset (300,895 SNPs) of 2919 individuals from Spain, Europe, north Africa[30] and sub-Saharan Africa[31] (Methods). We used fineSTRUCTURE to identify 29 non-Iberian donor groups (Methods). We extended the fineSTRUCTURE model to re-cluster individuals within Iberia, now based only on their levels of ancestry sharing across these 29 groups (Methods). These clusters capture the impact of migration into and across Spain, removing the effects of simple isolation events.

Using this approach we inferred six distinct clusters within Iberia (Fig. 5a), many fewer than in the Spain-only analysis (Fig. 1a), implying that much of the fine-scale structure seen within Spain is a result of regional genetic isolation. The six clusters still associate with geographical regions, predominantly in the east–west direction rather than north–south. Notably, the extensive sub-structure in Pontevedra disappears, and indeed these individuals now co-cluster with Portuguese individuals. Therefore, the extensive fine-scale structure in Galicia is most likely explained by local drift effects. In contrast, a distinct cluster still occurs within the Basque region. This indicates that alongside regional isolation, distinctive levels of ancestry sharing with non-Spanish groups contribute to fine-scale structure in this region.

To characterise the genetic make-up of these six Iberian clusters we estimated their ancestry profiles: we fitted each cluster as a mixture of (potentially) all 29 donor groups to approximate the unknown ancestral groups that actually contributed to modern-day Iberian individuals (Methods). This approach

accounts for the stochasticity in ancestral relationships along the genome and was previously shown to be informative in the context of the British Isles[6]. Only six of the 29 donor groups show a contribution >1% in Iberia, and all are located in Western and Southern Europe, and north-west Africa (Fig. 6). For all six Iberian clusters the largest contribution comes from France (63–91%), with smaller contributions that relate to present-day Italian (5–17%) and Irish (2–5%) groups. With the exception of the Basque cluster, these three donor groups contribute proportionally similar amounts throughout Iberia, so probably represent ancient ancestry components rather than recent migration. In contrast, north Moroccan ancestry shows strong regional variation (Fig. 5c, Methods). See Supplementary Note 7 for a fuller discussion of the ancestry profiles.

To distinguish between possible scenarios that could produce these patterns, we applied the GLOBETROTTER method[25] to each of our six clusters (Methods). GLOBETROTTER infers dates of admixture and the make-up of the source populations, and tests whether admixture patterns are consistent with a simple mixing of two groups at a single time in the past, compared to more complex alternative models. GLOBETROTTER found strong evidence ($p < 0.01$) of admixture for all six clusters (Methods; Supplementary Table 3a). For all six clusters, an extremely similar event was inferred (Fig. 5b), in a tight time-range of 860–1120 CE, and with similar source groups, present in varying proportions (4–10% for the minor group). The major source was inferred to contain almost exclusively European donor groups, and the minor source is made up of mainly north-west African donor groups, including Western Sahara, and to a lesser extent west Africans (YRI), consistent with the overall ancestry profiles. The 'Portugal-Andalucia' cluster shows the greatest YRI contribution, and also shows some evidence of a second admixture date, with a more recent event involving only sub-Saharan-African-like and European-like source groups (see Supplementary Figure 7 and Supplementary Note 8.2). This indicates a recent pulse of sub-Saharan African DNA, independent of the north African component. For the other five clusters, the dates are more precise than any previous estimate that used north African haplotypes in the analysis[20,25,26]. In our results any one 95% confidence interval (CI) spans no more than 11 generations (~300 years) and all confidence intervals combined span less than 14 generations (< 400 years).

GLOBETROTTER shows a subtle preference for Western Sahara as a source of north African DNA, as opposed to north Morocco. This might be explained if modern-day north Moroccan haplotypes are more similar to present-day Spanish individuals than the historical source population was. Indeed, a mixture analysis we performed of the north Moroccan group itself (Supplementary Figure 4; Methods) shows that this group has a non-trivial proportion of European-like ancestry while the

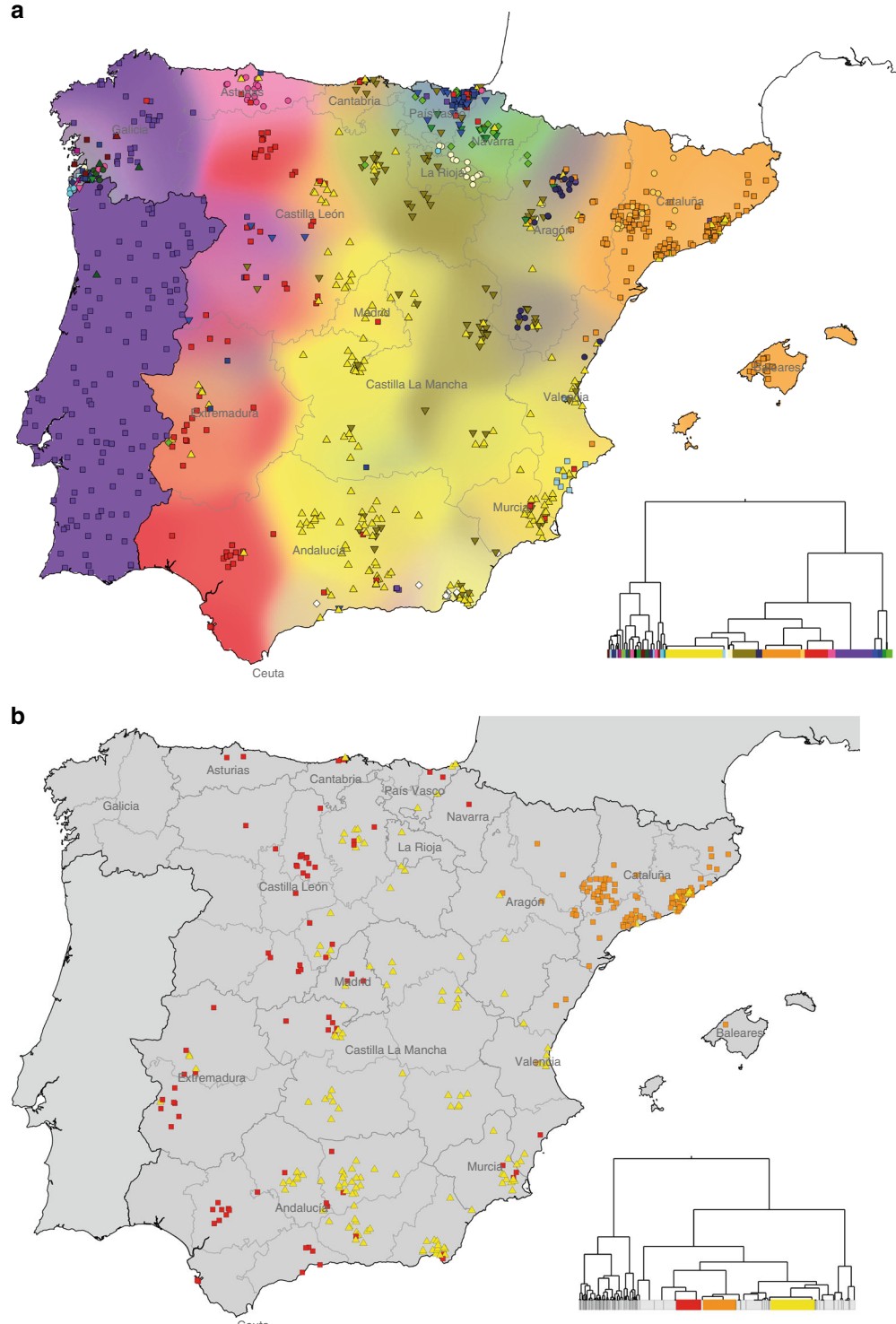

**Fig. 2** Clustering analysis including Portuguese individuals; and large clusters at the bottom of the tree. **a** This map and tree show clusters inferred by fineSTRUCTURE (analysis B) that included data from Portuguese individuals but using a smaller set of SNPs (Methods). As in Fig. 1b we show the level of tree such that all clusters contain at least 15 individuals (39 clusters). Points representing 843 individuals are shown on this map but, as with analysis A, data for all Portuguese and Spanish individuals (1530) were used in the inference. Positions of points and background colours are determined using the same procedure as for Fig. 1b (Methods), with the exception of Portugal. No fine-scale geographic information was available for these individuals, so we placed them randomly within the boundaries of Portugal and show a single background colour. **b** This map shows geographic spread of the three large clusters that remain at the bottom of the tree inferred in the Spain-only fineSTRUCTURE analysis (see main text; Fig. 1a). These clusters each contain more than 100 individuals out of the full set of 1413. The accompanying tree highlights the three clusters within the full tree structure. The width of the coloured rectangles is proportional to the number of individuals belonging to each cluster (yellow = 222; orange = 165; red = 123)

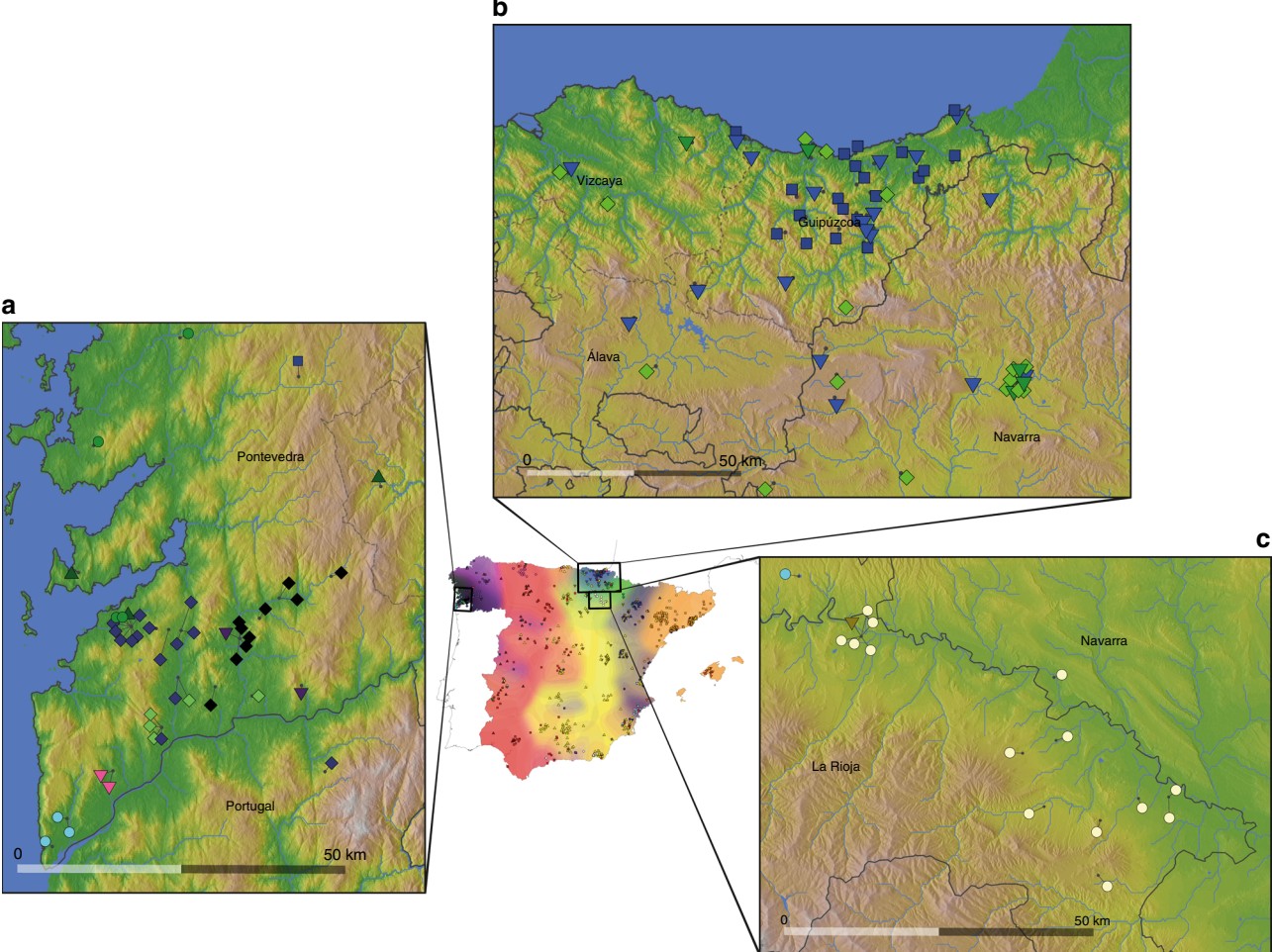

**Fig. 3** Ultra-fine-scale genetic structure within Spain. Points representing individuals are placed on each of the magnified maps and coloured as described in Fig. 1, with short dark lines pointing to their precise locations (the average birthplace of their grandparents). The three magnified maps show local elevation, rivers, and water bodies, as well as borders of autonomous communities (solid black lines) and provinces (dashed lines and text). **a** Locations of individuals (44) within the genetic clusters centred in Galicia. Note that we show this region at a higher level of the tree (14) as the lower level yields clusters with fewer than three individuals with fine-scale geographic location data. **b** Locations of individuals (60) within the clusters centred in the Basque-speaking regions of País Vasco (Basque Country) and Navarra. For visual clarity we only show the individuals that are within the clade coloured blue and green in Fig. 1. This clade makes up the majority of all individuals located in this region, and a majority of this clade is located in this region (60 of 64 with geographic data). **c** Locations of individuals (16) who almost all comprise a single cluster exclusive to a ~50-km-wide region along the banks of the River Ebro in La Rioja, just south of País Vasco and Navarra

Western Sahara donor group has none. Previous work showed similar results[30]. If this European-like ancestry had arrived more recently than the detected admixture event, the north Moroccan donor group would be a poor proxy for the historical source population and GLOBETROTTER would use a better alternative. Since GLOBETROTTER detects admixture based on the DNA received by the target population (Iberia) this would not affect the date estimates[25].

Our earlier results imply the incorporation of Basque-like DNA elsewhere within Spain. We next incorporated the clade labelled 'Basque1' as a potential donor group, to characterise and date this event (Methods). Ancestry profiles show contributions of Basque-like DNA (Fig. 5d) highest in places immediately surrounding the main location of the Basque donor group (País Vasco), and much higher southwards than to the east and west. GLOBETROTTER yields congruent results, and inferred dates for the arrival of Basque-like DNA in the range 1190–1514 CE, more recent than the north-west African influx (Fig. 7).

## Discussion

Our observation that genetic differences are small in the north–south direction within Spain, and evidence of gene flow preferentially in this direction, are most straightforward to interpret in the light of historical information regarding the *Reconquista*, during which Christian-controlled territory in the north moved gradually southwards from the mid-8th Century, following the Muslim conquest of Iberia (711CE). By 1249, almost all of Iberia was under Christian rule, and the Battle of Granada in 1492 marks the end of Muslim rule in Iberia. There is historical evidence of migration of peoples from the northern Christian kingdoms into newly conquered regions during the *Reconquista*[10,32]. The east–west boundaries of the clusters we see in the north of Spain correspond closely to the regions of broad linguistic differences in the Christian-ruled north, which date back to at least the first 200 years of Muslim rule (Fig. 1c), and we date the southwards movement of Basque DNA later within the *Reconquista* itself. Thus, it appears that present-day population

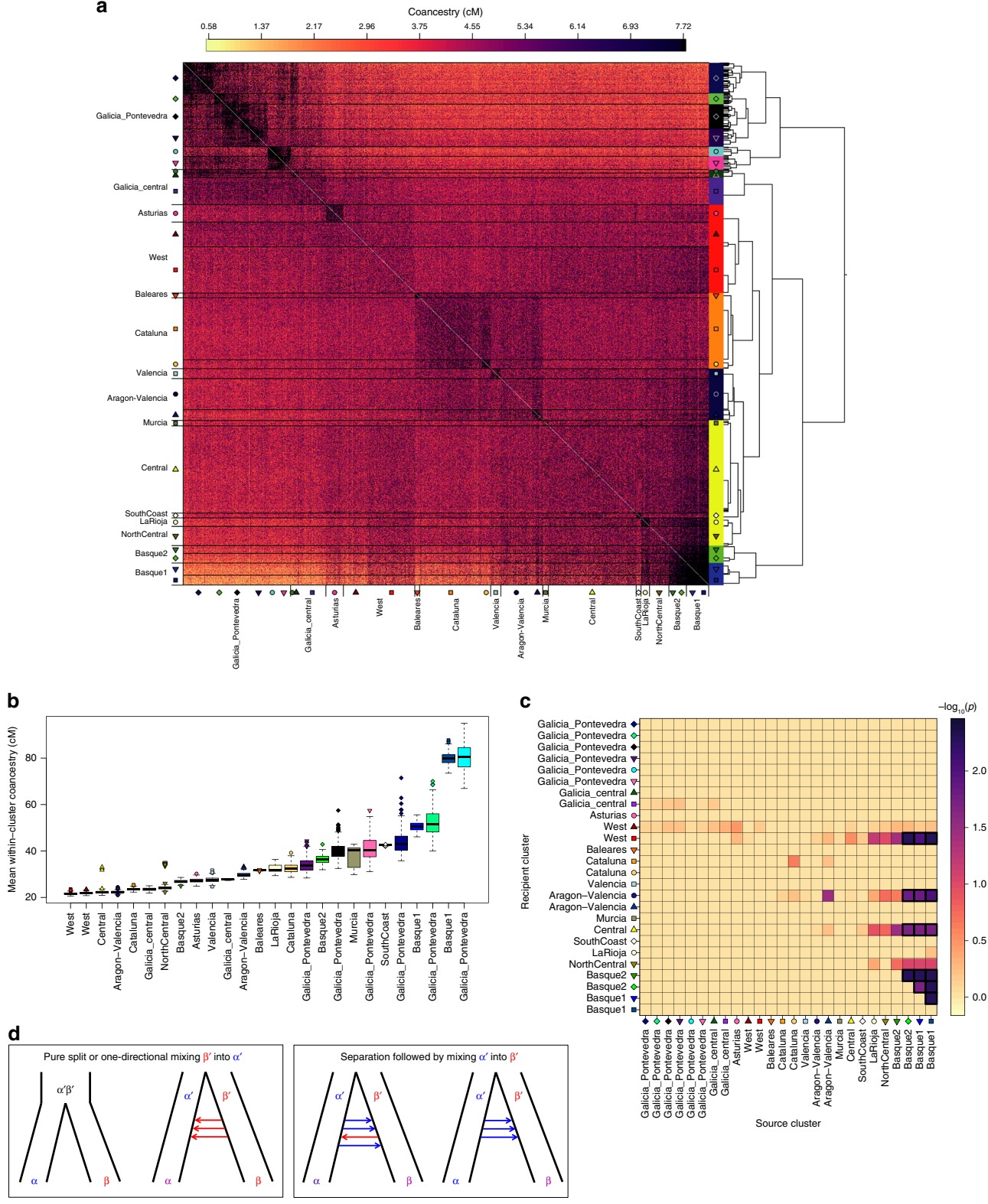

structure within Spain is shaped by population movements within this key period.

We also detect a genetic footprint of the Muslim conquest, and subsequent centuries of Muslim rule. Following the arrival of an estimated 30,000 combatants[33], a civilian migration of unknown

numbers of people occurred, thought to be mainly Berbers from north Morocco and settling in many parts of the peninsula[33]. Our analysis confirms and refines previous findings[11,20,26] of a substantial and regionally varying genetic impact, narrowing to a period spanning < 400 years. Crucially, unlike previous genetic

**Fig. 4** Estimates of shared ancestry between Spanish individuals and across fineSTRUCTURE clusters. **a** Matrix of coancestry values used in cluster inference. Each of 1413 individuals is represented as a row, where each element is the coancestry (in cM) shared with each of the other individuals (see Methods for the definition of coancestry). In order to visualise the bulk of the variation, values equal to or above the 90th percentile (7.7 cM) are coloured black. The tree is as shown in Fig. 1a, and the horizontal black lines demarcate the clusters at the lower level of the tree, and labelled with points. **b** The distribution of the mean coancestry between individuals in the same cluster for 200 bootstrap resamples (Methods). Clusters are ordered by their median value, and coloured/labelled according to those shown in **a**. One cluster (part of the clade labelled 'Galicia_central') was excluded from this analysis as it only contains 9 individuals. **c** Evidence for excess of coancestry with a source cluster compared to within-cluster coancestry. Each row of this matrix is a cluster inferred in the fineSTRUCTURE analysis as labelled in **a**. For each recipient cluster (rows) we tested whether the mean coancestry among individuals within the recipient cluster is smaller than their mean coancestry with individuals in each of the other clusters (columns). Each element is coloured according to $-\log10(p)$, where $p$-values are based on 200 bootstrap resamples using the same sample size (13 individuals) for all clusters (Methods). Dark borders indicate source-recipient pairs with a $p$-value < 0.02 (not Bonferroni corrected). **d** Illustration of demographic scenarios leading to high coancestry between two different clusters. The symbols α and β represent clusters of individuals today, and α' and β' represent their ancestral populations. Arrows represent mixing of one ancestral population into the other at some time (or times) in the past. In the left two scenarios individuals in β will have—on average—higher coancestry with each other than with individuals in α. In the right two scenarios it is possible for individuals in β to have higher coancestry with individuals in α than with each other (see Supplementary Note 5 for a fuller discussion)

studies of admixture in Iberia[11,24,26], we avoid strong assumptions about the genetic make-up of the historical admixing groups. Instead of specifying in advance the modern-day sources that we assume best represent the historical populations that came together in the past, we infer the best mixture of modern-day populations from a large set of possible groups. Our GLOBETROTTER results suggest that amongst the six potential African populations in our study, the best match to the predominant group involved in the actual admixture event is northwest African. Moreover, admixture mainly, and perhaps almost exclusively, occurred within the earlier half of the period of Muslim rule (Fig. 5b). Within Spain, north African ancestry occurs in all groups, although levels are low in the Basque region and in a region corresponding closely to the 14th-century Crown of Aragon (compare Figs 1c, 5c). Therefore, although genetically distinct[22,23], north African-like ancestry in the Basque region could be explained through genetic interactions between the Basque groups and other parts of Spain within the past 1300 years.

Perhaps surprisingly, north African ancestry does not reflect proximity to north Africa, or even regions under more extended Muslim control. The highest amounts of north African ancestry found within Iberia are in the west (11%) including in Galicia, despite the fact that the region of Galicia as it is defined today (north of the Miño river), was never under Muslim rule[34] and Berber settlements north of the Douro river were abandoned by 741. This observation is consistent with previous work using Y-chromosome data[11]. We speculate that the pattern we see is driven by later internal migratory flows, such as between Portugal and Galicia, and this would also explain why Galicia and Portugal show indistinguishable ancestry sharing with non-Spanish groups more generally. Alternatively, it might be that these patterns reflect regional differences in patterns of settlement and integration with local peoples of north African immigrants themselves, or varying extents of the large-scale expulsion of Muslim people, which occurred post-*Reconquista* and especially in towns and cities[10,32].

We show that population structure exists at ultra-fine scales in Galicia (Fig. 3a), particularly in the province of Pontevedra, with some clusters having geographic ranges of less than 10 Km (root-mean-square distance from cluster centroid). To our knowledge, these results represent the finest scales over which such structure has yet been observed in humans. Previously, it has been shown that by jointly analysing people from a priori defined sampling locations, subtle differences in group averages at certain genomic loci can be observed at fine geographic scales. For example, subtle differences in blood group frequencies have been observed among villages in Italy's 70-km-long Parma Valley[35]. Our results go

beyond this, showing that by leveraging information genome-wide, it is possible to detect subtle genetic structure at fine geographic scales, without utilizing prior geographic information. It will be interesting to identify whether (and if so which) other parts of the world show similar patterns. Pairs of individuals within these clusters show high levels of coancestry relative to the rest of Spain (Fig. 4b). In contrast, when we only consider their patterns of coancestry with non-Iberian groups this structure disappears: individuals from Pontevedra are indistinguishable from those from Portugal and other parts of western Spain (Fig. 5a). Therefore, the very strong population stratification observed in Galicia can most easily be explained through very recent geographic isolation, occurring subsequent to major migrations into the region (see Supplementary Note 4 for further discussion). It is worth noting that differences in the amount of population structure observed in different regions may be sample-dependent (see Supplementary Note 2.3 for discussion). In principle, if sampling is differentially biased towards, for example, rural verses urban areas in different parts of Spain it could potentially lead to differences in detected patterns of structure. This might mask structure in some regions, but crucially, our approach would not be able to find structure if it was not there.

Overall, the pattern of genetic differentiation we observe in Spain reflects the linguistic and geopolitical boundaries present around the end of the time of Muslim rule in Spain, suggesting this period has had a significant and long-term impact on the genetic structure observed in modern Spain, over 500 years later. In the case of the UK, similar geopolitical correspondence was seen, but to a different period in the past (around 600 CE)[6]. Noticeably, in these two cases, country-specific historical events rather than geographic barriers seem to drive overall patterns of population structure.

It appears that within-country population structure occurs across the world[6,36], and in this study is observed down to scales of <10 km. Such strong, localised genetic drift predicts the existence of geographically localised rare mutations, including pathogenic ones. For example, cases of a specific form of inherited ataxia (SCA36) cluster within a specific region of Galicia[37]. Our results imply this phenomenon will tend to happen in specific areas, and can arise even where population density is high and in the absence of obvious strong geographic barriers, such as Galicia in the case of Spain.

## Methods

**Data and quality control**. For the Spain-only analysis we used genotype data that was originally collected and typed for a colorectal cancer GWAS[38]. Biological samples were sourced from a variety of hospitals across Spain as well as the Spanish

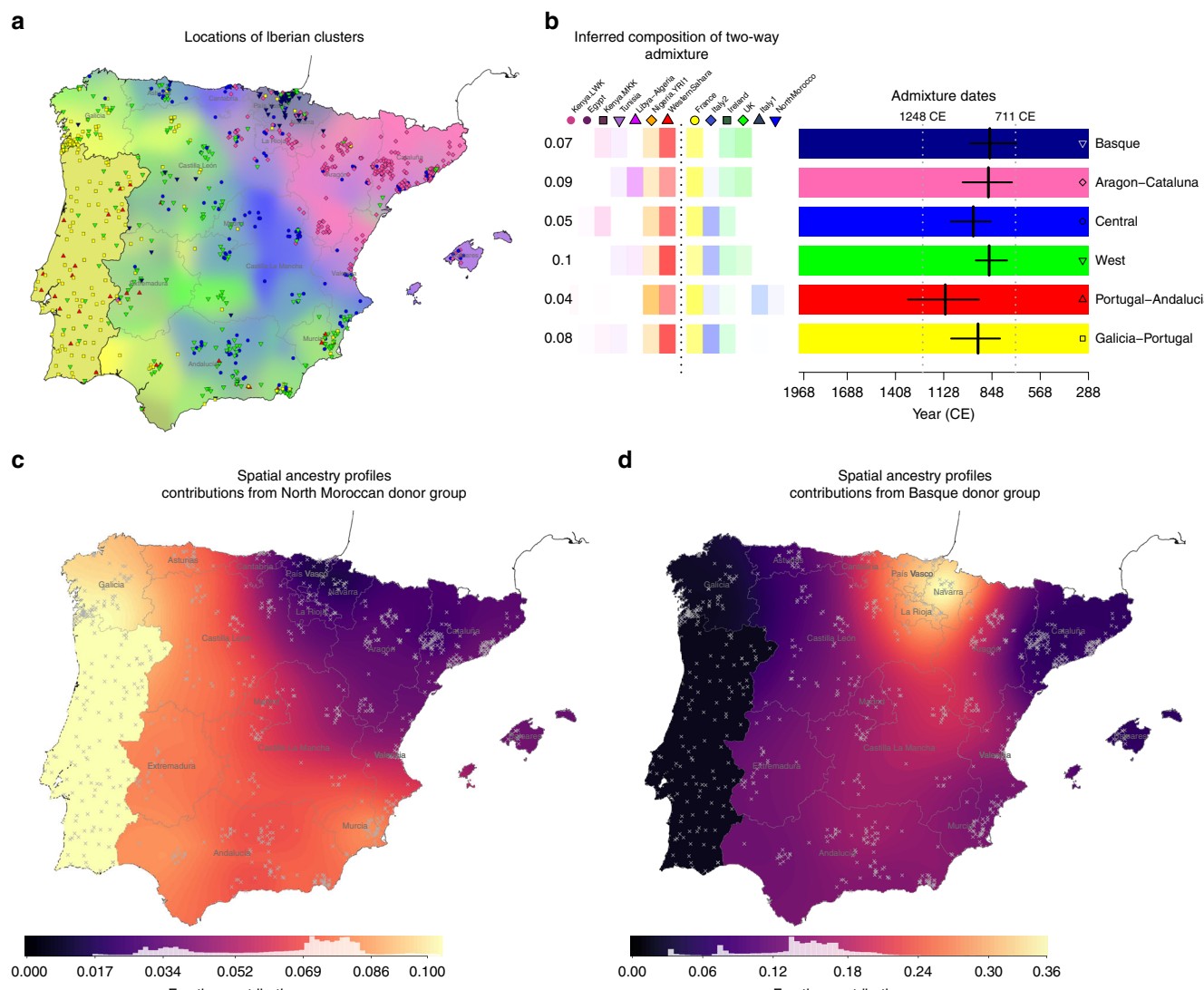

**Fig. 5** Characterising genetic contributions to Iberia. **a** Geographic distribution of 843 Iberian individuals grouped into six clusters based on haplotype sharing with external populations (Methods). More individuals (1530) were used in the inference, but only those with adequate geographic data are shown on the map. Background colours and the positions of points on the map are determined using the same procedure as for Fig. 1b, with the exception of individuals of Portuguese origin. No fine-scale geographic information was available for these individuals, so we placed them randomly within the boundaries of Portugal and show a single background colour (Methods). **b** Admixture dates and mix of admixing groups in single-date, two-way admixture events, as inferred using GLOBETROTTER (n = 541 individuals). On the left are the donor groups inferred to best represent the two ancestral populations involved in the admixture event (separated by a dashed line), along with the inferred admixture proportions of the smaller side (for donor groups contributing at least 1%). Estimated dates and 95% bootstrap intervals are shown on the right, for each target Iberian cluster as shown in **a**. The white vertical dashed lines show the time of the initial Muslim conquest (711 CE) and the Siege of Seville (1248 CE), between which around half (or more) of Iberia was under Muslim rule. The admixture dates assume a 28-year generation time, and a current generation date of 1940 (the approximate average birth-year of this cohort). Detailed results of this GLOBETROTTER analysis are tabulated in Supplementary Tables 3a and 4. **c**, **d** We estimated ancestry profiles for each point on a fine spatial grid across Spain (Methods). The background colour shows the fraction contributed from a particular donor group, as defined by the scale bar. Grey crosses show the locations of the Iberian individuals used in the estimation: 843 in **c**, 793 in map **d**. Map **c** shows the fraction contributed from the donor group 'NorthMorocco'. Map **d** shows the fraction contributed from the donor group 'Basque1', which we defined based on the Spain-only fineSTRUCTURE analysis (Fig. 1a). Maps for other donor groups are shown in Supplementary Figure 5

National DNA Bank. All samples were assayed together by Affymetrix (now Thermo Fisher Scientific) in the same facility. See ref. [38] for full details of sample collection and genotype calling. We used both cases and controls in Phase I of the GWAS study, which totalled 1548 individuals prior to quality control. Individuals from all 17 of Spain's autonomous communities are represented in this dataset, but the Spanish territories of Melilla, Ceuta, and the Canary Islands are excluded from analyses involving geographic labelling due to limited sampling in these areas (four samples).

After applying a series of quality control filters (Supplementary Note 1.1) we phased the genotype data using SHAPEIT v2[39] with a reference panel and

recombination map from Phase I 1000 Genomes[40]. Close relatives (kinship coefficient > 0.1), and individuals with evidence of recent, non-Spanish ancestry were included in the phasing, but excluded from the fineSTRUCTURE analysis. This procedure resulted in 1413 samples and 693,092 SNPs for our main analysis.

For all analyses involving individuals from outside of Spain we combined four sources of genotype data: European samples from POPRES[41], north African samples from ref. [30], sub-Saharan African samples from Hapmap Phase 3[31], and the Spanish samples described above. The POPRES and north African samples were typed on the Affymetrix 500K array, which overlaps substantially with the

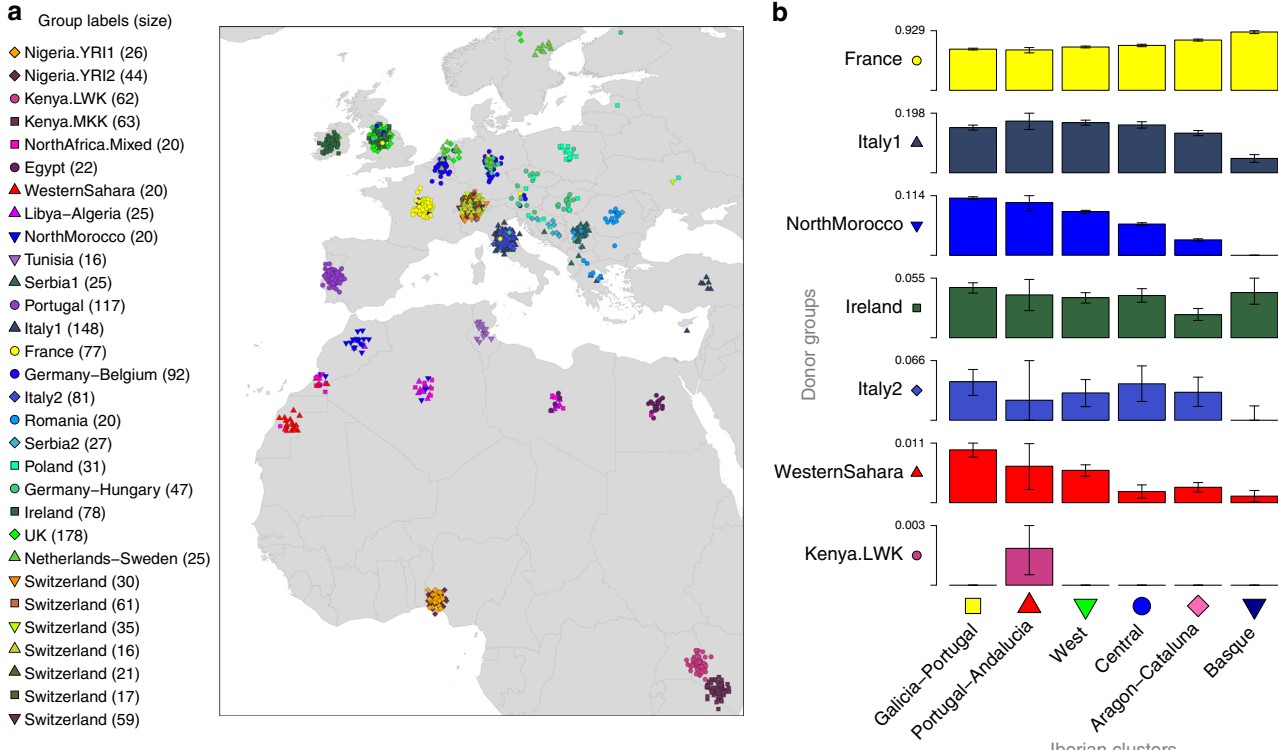

**Fig. 6** Locations of donor groups and ancestry profiles of Iberian clusters. **a** Locations of individuals ($n = 1503$) within 30 non-Spanish genetic groups inferred using fineSTRUCTURE (Methods). Each point represents an individual, placed at their country-level location of origin, and coloured according to their inferred genetic group. Individuals from the same location (country) have been randomly jittered for visual clarity. Names are assigned to clusters based on where the majority of the individuals in the clusters are located. Where a cluster was split more evenly across two regions, a double-barrel name is used. All groups shown here, except 'Portugal', were used as donor groups in the analyses of Iberia. **b** Each column shows the ancestry profile for each of the inferred clusters shown in Fig. 5a. The heights of the bars show the proportion of each cluster's ancestry which is best represented by that of the labelled non-Iberian donor group (Methods). Note that each row has a different y-axis range for visibility of the smaller components. Error bars show the range of the inner 95% of 1000 bootstrap resamples (Methods), and donor groups are only shown if at least one cluster has a range not including zero and a point estimate >0.001. The exact values plotted here and cluster sample sizes are in Supplementary Table 1

Affymetrix 6.0 array. We first merged these data and then applied quality filters to the combined dataset of 6617 individuals, which we then phased altogether. After excluding related individuals, those with self-reported mixed ancestry, and sub-sampling some heavily sampled regions (e.g. Switzerland) the dataset comprised 2919 individuals and 300,895 SNPs, which we used in our joint analyses. See Supplementary Note 1.2 for full details of quality control and phasing.

**Clustering using fineSTRUCTURE**. We inferred clusters of individuals based on genetic data only by applying the fineSTRUCTURE method[28], which uses a model-based approach to cluster individuals with similar patterns of shared ancestry. Within the fineSTRUCTURE framework, shared ancestry is measured as the total amount of the genome (in centiMorgans (cM)) for which individual $i$ shares a common ancestor with individual $j$, more recently than all the other individuals in the sample. This is estimated for each pair of individuals $i$ and $j$, defining a square matrix referred to as the coancestry matrix (e.g. Fig. 4a). This matrix is then used to cluster individuals into groups with similar patterns of coancestry, i.e. similar rows and columns in the matrix. We applied fineSTRUCTURE using the procedure recommended by the authors, except in one aspect: to measure shared ancestry (coancestry) we used the total amount of genome (in cM) for which individual $i$ shares a common ancestor with individual $j$, more recently than all the other individuals in the sample. The software default coancestry measure is the number of contiguous segments (chunks) rather than the total amount of genome, but we found the alternative measure to be more robust to artefacts such as genotyping error (Supplementary Figure 2; Supplementary Note 2.1).

Using the coancestry matrix, fineSTRUCTURE applies a Markov chain Monte Carlo (MCMC) procedure to find a high posterior probability partition of individuals into a set of clusters. The number of clusters is not specified in advance, but rather estimated under the fineSTRUCTURE probability model. Having found a set of clusters, fineSTRUCTURE then infers a hierarchical tree by successively merging pairs of clusters whose merging gives the smallest decrease to the posterior probability (of the merged partition) among all possible pairwise merges.

We ran several fineSTRUCTURE analyses using phased haplotypes from the following sets of individuals:

(A)  Spanish individuals
(B)  Spanish and Portuguese individuals
(C)  Non-Spanish individuals

In all cases we used CHROMOPAINTER software (v2)[28] to estimate the coancestry matrix, followed by fineSTRUCTURE's clustering and tree-building procudures. We used a previous successful application of fineSTRUCTURE[6] as a guide for the number of iterations in the MCMC, and other required parameters (see Supplementary Note 2 for full details). We also checked that the MCMC samples were largely independent of the algorithm's initial position by visually comparing the results of two independent runs starting from different random seeds. Good correspondence in the pairwise coincidence matrices of the two runs indicates convergence of the MCMC samples to the posterior distribution[28]. See, for example, Supplementary Figure 1a showing the two independent runs for analysis A. Without loss of generality, we used the first of these two runs in our main analysis.

**The statistical uncertainty of cluster assignments**. For analysis A, we measured uncertainty in the assignment of individuals to clusters by using a procedure described formally in ref. [6], which uses the information from multiple samples of the fineSTRUCTURE MCMC. Informally, the procedure measures the overlap between a cluster $k$, and individual $i$'s assigned cluster in each of the MCMC samples within a single fineSTRUCTURE run. This can take values between 0 and 1, and sums to 1 across all clusters for a given individual. It provides a measure of certainty about the assignment of individual $i$ to each cluster $k$ in the final set of clusters. The cluster assignment certainty for an individual is the value corresponding to the final cluster assignment, and will be close to 1 if they are assigned to a cluster with largely the same set of individuals in each MCMC sample. This measure can be obtained for different layers of the hierarchical tree by summing the values for the clusters that merge at each layer.

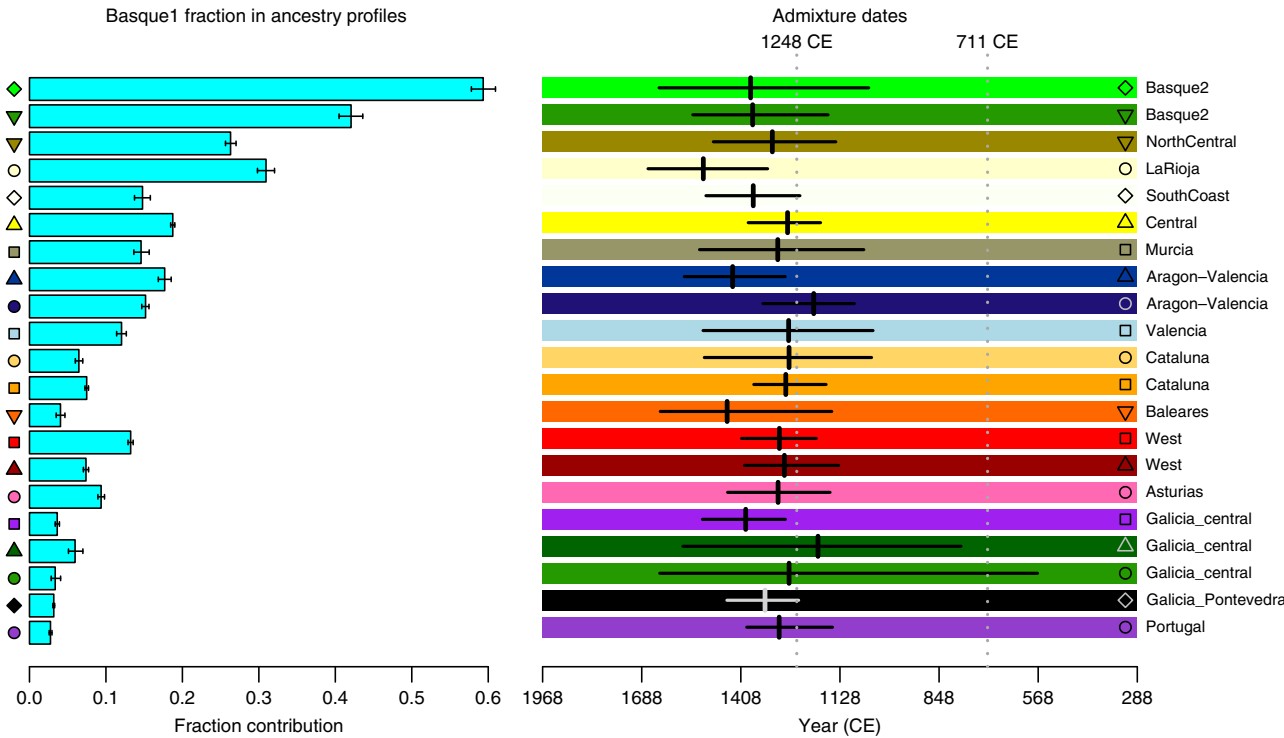

**Fig. 7** Variation and timing of Basque-like genetic contributions in Iberia. Fraction contributions from the Basque-like donor group in ancestry profiles (Methods) and Basque-like admixture dates (GLOBETROTTER) for each cluster inferred in the Spain-only analysis (as shown in Fig. 1a) plus Portugal. The clade labelled 'Galicia_Pontevedra' in Fig. 1a was combined into one group for this analysis. The admixture dates are for a two-way admixture event involving a Basque-like side and an European-like side, and shown with 95% bootstrap intervals (Methods). The dates shown assume a 28-year generation time, and a current generation date of 1940. Detailed results of this GLOBETROTTER analysis are in Supplementary Table 3b

For fineSTRUCTURE analysis A, cluster assignments at higher levels of the hierarchical tree are typically more certain than lower-level clusters. At the broader level of the tree shown (14 background colours in Fig. 1b) 94% of individuals have a cluster assignment certainty >0.7; and at the finer level (shown with points in Fig. 1b), this level of certainty is reached by 89% of individuals (see Supplementary Figure 1b). Furthermore, the clusters with the highest certainty tend to be those with greater geographic localization, e.g. those labelled 'LaRioja' and 'Baleares' (Fig. 1).

**Selection of levels of the Spanish tree to analyse**. In the fineSTRUCTURE analysis A 145 clusters were inferred. In order to examine properties of these clusters, such as their geographical locations, we focused on two different levels of the tree, thus highlighting both broad-scale and fine-scale structure. There is no right level of the tree to pick, but we chose them based on the sizes of the clusters. To examine broad-scale structure we chose a level (14 clusters) such that all clusters were larger than size 20. Recall that moving up the tree, at each level a pair of clusters is merged. At the base of the tree, newly separated groups typically contain few (<15) individuals, with higher clustering uncertainty. To avoid the presence of such minor clusters, we traversed up the tree until the first time a merge occurred between two larger clusters (>15 individuals). This occurs at the level of 49 clusters. However, at this level over half (28) of the clusters are within the clade involving individuals mostly from south-west Galicia (labelled 'Galicia_Pontevedra' in Fig. 1a), and for many of these clusters fine-scale geographic information was only available for one or two individuals, and/or the cluster contained fewer than 15 individuals. Therefore, to aid visualization in Fig. 1b and Fig. 3a we only show the clusters at the higher level of the tree (level 14) for this clade, although the full tree is still shown in Fig. 1a.

**Map-based data visualisation**. Geographic information was available for individuals in the Spanish cohort, along with their age at collection, sex, and genotyping plate (controls only) and batch used in genotype calling. The geographic information includes region of origin (autonomous community) for all individuals; for 959 individuals (68%) the birthplace (town) of at least one grandparent was available, and for 883 of these the birthplace of all four grandparents was available. We assigned each individual to a geographic coordinate by matching the text (e.g. 'Barcelona') to a municipal region as defined by the Spanish Statistical Office (www.ine.es, 2014) and coding them to the centre of the matching region. Some locations were not themselves a municipality, so we coded these individuals to the centre of the nearest municipality, identified by using Google Maps. However, in

the fineSTRUCTURE analysis we clustered all individuals, i.e. also included those individuals for whom the exact birthplaces of their grandparents was not known, but these are not used in interpreting the spatial distribution of the inferred genetic structure.

In figures showing a map of Spain (e.g. Fig. 1b) each individual is represented by a point placed at the average coordinate (centroid) of their grandparents' birthplaces (coordinates were derived as described above). To minimise the effects of very recent (20th Century) migration within Spain, only those individuals (726) with all four grandparents born within 80 km of each other are shown on the maps. See Supplementary Table 6 for the effect of this filtering in different regions of Spain. Where many individuals have the same coordinate, such as in Barcelona, points have been randomly shifted, by no more than 24 km, to aid visualisation. To visually represent the discrete assignment of individuals to clusters by fineSTRUCTURE, the members of each cluster are represented using the same colour and symbol. We also coloured the background of the maps using a Gaussian kernel smoothing procedure on a regular grid of 3-km-wide squares across Spain. Informally, each square in the grid is coloured according to the relative contributions of each cluster, where contributions are measured by Gaussian densities centred on the location of each individual. See Supplementary Note 3 for a formal description.

The linguistic maps in Fig. 1c are based on scanned images of previously published work (Baldinger, K. La Formación de los Dominios Lingüísticos en la Península Ibérica, Spain: Gredos, 1963, maps 6-9, 10[29]), which we adapted in the following ways using Adobe Illustrator. Colours were added to help distinguish the different linguistic groups and the political borders (from Map 10 by Baldinger[29]) were overlaid onto the right-most map.

**Signals of drift and admixture in the coancestry matrix**. Recall that coancestry (as we have used it) measures the amount of genome (in cM) for which an individual $i$ shares its most recent common ancestor with another individual $j$, out of all the individuals in the sample. Properties of this matrix are informative of patterns of drift and admixture within and across clusters inferred by fineSTRUCTURE. Specifically, excess coancestry between individuals in the same cluster (within-cluster coancestry) is a natural measure of genetic drift of that cluster relative to all the other clusters[28]. In general, individuals are often observed to have the highest levels of coancestry with other individuals in their assigned cluster. This is not a constraint of the fineSTRUCTURE model; rather it is because if two individuals have similar patterns of shared ancestry, they are naturally also likely have more recent shared ancestry between them. However, it is possible for this not

to be the case, and this is informative of admixture (see Supplementary Note 5 for details).

We looked for such signals in the case of Spain by testing whether a cluster (inferred by fineSTRUCTURE) has a within-cluster coancestry that is, on average, smaller than its coancestry with another cluster. We used the 26 clusters (indicated with symbols on the axes in Fig. 4a), which contained at least 13 individuals. To avoid potential bias due to uneven sizes of the clusters, we estimated within-cluster coancestry levels by randomly sub-sampling (without replacement, as coancestry is only defined between two different individuals) each of the 26 clusters such that there were of equal size (13). We re-computed the coancestry matrix using CHROMOPAINTER, and the same set of parameters as in fineSTRUCTURE analysis A, but using this smaller subset. We repeated this 200 times and used these resamples to compare coancestry levels across clusters. For each resample and each cluster we computed the mean of the coancestry values within that cluster (excluding zeros on the diagonals), and with each of the other clusters. We then computed a p-value using the number of resamples ($S$) for which the mean within-cluster coancestry is smaller than the mean coancestry with each of the other clusters. That is: $= \frac{S+1}{201}$. Results are shown in Fig. 4b, c.

**Principal components and $F_{ST}$.** For both PCA and $F_{ST}$ analyses we used a set of 143,599 LD-pruned SNPs ($r^2 < 0.2$) by applying the '--indep-pairwise r2' command in PLINK (v1.7)[42], and excluded regions of long-range LD derived from[43]. We computed principal components of the genotype calls using the software Shellfish[44], and $F_{ST}$ between different groups of individuals using EIGENSOFT (v5.0.1)[45]. We conducted two $F_{ST}$ analyses: one using autonomous community as group labels, and the other using clusters inferred by fineSTRUCTURE as group labels.

In the $F_{ST}$ analysis using autonomous community as group labels the strongest differentiation (but still weak, at 0.002) is between the Basque-speaking regions and all other regions, and between Galicia and other regions (Supplementary Figure 9b). The principal components analysis revealed a similar pattern, separating the same regions (Supplementary Figure 9a). The range of pairwise $F_{ST}$ values is higher when grouping individuals by the clusters inferred by fineSTRUCTURE (0–0.008) (Supplementary Figure 9c). This highlights fineSTRUCTURE's ability to find clusters of individuals who share genetic drift, and reveals two highly drifted clusters within Galicia and the Basque region.

**Defining donor groups.** In order to define a set of donor groups using the combined European, north African, and sub-Saharan African individuals, we applied fineSTRUCTURE as described above in several rounds, each using the following sets of individuals:

(CI) All individuals combined (excluding Spanish but including Portuguese)
(CII) Individuals from north Africa only
(CIII) Individuals from Europe only

In analysis CI, fineSTRUCTURE cleanly split the three main groups corresponding to Europe, north Africa, and sub-Saharan Africa, as well as inferring finer sub-structure. However, in order to maximise the power to detect finer scale structure[28], we obtained fineSTRUCTURE results for the north African and European groups independently. That is, using coancestry matrices that only allow copying within each set of individuals in CII and CIII, respectively. We then defined a set of donor groups based on the clusters and hierarchical trees inferred by fineSTRUCTURE in these three analyses. We considered the following factors in defining donor groups. Ideally, each donor group would contain about the same number of samples, and not be too small. Donor groups should also be relatively homogeneous with respect to their shared ancestry with the population of interest (in this case Iberia), although could be heterogeneous within themselves. We therefore prioritised donor group size over capturing finer scale structure that might exist within donor groups themselves. See Supplementary Note 6 for full details. Our procedure resulted in a total of 29 donor groups with median size 30, and minimum size 16, totalling 1386 individuals (not including Portuguese). Their locations are shown in Fig. 6a. Labels of the inferred groups are based on the sampling locations of most of the individuals in a given group. In some cases the majority of individuals were split across two locations, and this is indicated by a multi-region label (e.g. Germany–Hungary).

**Treatment of Portugal.** One cluster in the fineSTRUCTURE analysis CIII overlaps significantly (98%) with the individuals with grandparental origins in Portugal as reported by the data source (POPRES). For the purposes of the analyses in this chapter (and fineSTRUCTURE analysis B), this group of 117 individuals is referred to as 'Portugal' or 'Portuguese individuals' (e.g. in Fig. 6a). The strong genetic similarity between individuals from Portugal and Spanish individuals, especially those located in Galicia (Fig. 2a), means they are likely to share a similar admixture history, and including Portugal as a donor group would mask the signal from those shared events. We therefore excluded them from the set of donor groups and instead treated them in the same way as the Spanish individuals, bringing the total number of Iberian samples we analysed to 1530. This is analogous to the rationale for excluding Ireland as a donor group in the British Isles study[6].

**Clustering based on haplotype sharing with external groups.** Here we describe the method we used to infer clusters of Iberian individuals with distinct patterns of

haplotype sharing with external groups. We used the fineSTRUCTURE clustering algorithm but with a modified version of the input coancestry matrix. Specifically, we compute the coancestry (using CHROMOPAINTER) between each Iberian individual and each of the non-Iberian individuals, as described above, but only allowing Iberian individuals to copy from non-Iberian individuals. This results in a rectangular matrix, $X$, with $N$ rows and $M$ columns, where $N$ is the number of Iberian individuals and $M$ the number of non-Iberian individuals. We then constructed an $(N + M) \times (N + M)$ square matrix $C$, such that,

$$C = \begin{pmatrix} 0 & X \\ 0 & Y \end{pmatrix} \tag{1}$$

and matrix $Y$ contains zeros, except for each of the (block) diagonal entries corresponding to pairs of individuals within the same donor population $k$. These entries each take the value $g_k$, which is determined such that the mean of all the entries corresponding to donor population $k$ are the same for the sub-matrix $X$ as the sub-matrix $Y$. The zeros in the matrix $C$ have the effect of not allowing any copying from or to the Iberian individuals to contribute to the fineSTRUCTURE likelihood. We then run fineSTRUCTURE algorithm using the force file option (-F), where each 'continental' group is a donor group, thus only allowing splits and merges to take place among Iberian individuals. We used a c-factor of 0.0579, which was computed in the manner described in Supplementary Note 2.2, but using segments of DNA from a CHROMOPAINTER run where we only allowed Iberian individuals to copy from non-Iberian individuals.

**Estimating ancestry profiles.** We estimated ancestry profiles for each of the Iberian clusters using the procedure described previously[6]. Briefly, we use CHROMOPAINTER to compute a coancestry vector for each Iberian individual, where we only allow them to copy from haplotypes in the donor groups (as with matrix $X$ above), and then average the coancestry vectors within each cluster and donor group. For each Iberian cluster $i$, we then find a smaller set of donor groups which together (as a non-negative linear mixture whose coefficients sum to 1) best explains its cluster-averaged coancestry vector, $\bar{y}_i$ (see Supplementary Note 7.1 for full details). The vector of coefficients in the linear mixture is the ancestry profile for cluster $i$, and its elements sum to 1. Results for six Iberian clusters are shown in Fig. 6b. We also performed a complementary analysis where we treated each donor group in turn as $\bar{y}_i$, after removing their corresponding elements in the coancestry vectors and re-normalising (Supplementary Figure 4; Supplementary Note 7.1).

We measured uncertainty in these ancestry profiles by re-estimating the cluster-averaged coancestry vectors using a set of pseudo individuals. Each pseudo individual is formed by randomly selecting an individual in cluster $i$ for each chromosome, and summing the observed chromosome-level coancestry vectors across all chromosomes. We then compute 1000 such re-estimations and report the range of the inner 95% of the resulting bootstrap distribution.

**Spatially smoothed ancestry profiles.** The availability of fine-scale geographic information for many of the Spanish individuals allowed us to estimate the spatial distribution of shared ancestry (Fig. 5c, d; Supplementary Figure 5). Instead of averaging coancestry over individuals within a cluster, we average across geographic space using a Gaussian kernel smoothing method that varies the kernel band-width depending on the density of available data points (see Supplementary Note 7.2 for details). This gives a set of coancestry vectors, $\bar{y}_s$, for each grid-point $s$ in a fine spatial grid across Spain. We then compute ancestry profiles for each of the grid-points in the same way as for the Iberian clusters (described above), but setting $\bar{y}_i = \bar{y}_s$ instead of a cluster-averaged coancestry vector. We visualize the results by colouring each grid-point according to the value of its coefficient for a single donor population of interest (e.g. NorthMorocco in Fig. 5c). For any grid-point $s$ and individual $i$ located within the borders of Portugal we set all $\bar{y}_s$ to be the average coancestry vector across these individuals, because we have no fine-scale geographic information for them. This means in Portugal there is always one colour plotted.

**Estimating admixture dates and source populations.** We used the GLOBE-TROTTER algorithm to estimate dates, proportions and configurations of admixture events[25]. Briefly, GLOBETROTTER uses 'paintings' from the CHRO-MOPAINTER algorithm to construct coancestry curves. These curves measure the rate of decay of linkage disequilibrium with genomic distance, between sites with ancestry from a pair of source populations. The parameters of exponential functions fitted to these curves (decay rates and intercepts) are used to estimate admixture dates, admixture proportions, and the best fitting mix of modern-day groups that characterise the ancestral populations involved in an admixture event.

We conducted two analyses using GLOBETROTTER. The first (gtA) was designed to detect admixture event(s) in the history of Iberia that might involve any combination of non-Iberian source populations, without any prior assumptions on the nature of the event. The second analysis (gtB) was designed to detect only admixture event(s) involving a Basque-like source population, i.e. based on a prior hypothesis. In each case we defined a set of target groups within which to look for an admixture event; a set of donor groups, which we allow to be donors in the initial painting; and a set of surrogate populations, which we allowed

GLOBETROTTER to consider as components of any admixture event (Supplementary Table 2).

After identifying the presence of admixture based on criteria recommended by the authors, we next evaluated the evidence for more complex admixture events (e.g. multiple dates or more than two source populations). GLOBETROTTER automatically tests for these using a series of criteria based on how well the coancestry curves fit the models for different types of admixture scenarios[25]. Using GLOBETROTTER's automated criteria, in analysis gtB there was no evidence that a two-date admixture model fitted better than a one-date model. However, for some target populations in analysis gtA there was some evidence for a two-date admixture event, although the one-date event fit well in all cases (Supplementary Table 3a). Given the potentially complex nature of admixture in Iberia, we further evaluated the evidence for a two-date admixture event by considering the model fits for each coancestry curve separately (Supplementary Figure 7; Supplementary Note 8.2). Notably, only the coancestry curves involving a sub-Saharan African surrogate group fit better to a two-date admixture event. The improved fit for the curve for the sub-Saharan African surrogate group 'Nigeria.YRI1' is visually apparent in the coancestry curve shown in Supplementary Figure 7. We therefore consider the one-date admixture event to be a better fit overall, but that there is some evidence for a second event involving sub-Saharan African-like DNA mixing with European-like DNA, with the strongest evidence for this in the Iberian cluster, 'Portugal-Andalucia'. In the target groups where there is evidence of this, GLOBETROTTER infers dates in the range 1370–1700 CE (assuming a 28-year generation time).

**Ethics statement**. The collection of the Spanish genotype data was approved by the "Comité Ético de Investigación Clínica de Galicia", and each of the institutional review boards of the participating hospitals. All samples were obtained with written informed consent reviewed by the ethical board of the corresponding hospital, in accordance with the tenets of the Declaration of Helsinki. A previous analysis of these data was published in 2013[27]. All other data used in the paper were previously published and are publicly available for research use.

**Code availability**. Some specialist code used in the paper, which is not already publicly available, will be made available for academic research on request. We used the following versions of software: Birdsuite v1.4, KING v1.4, QCTOOL v1, SHAPEIT v2, PLINK v1.7, CHROMOPAINTER v2, fineSTRUCTURE v0.0.5, GLOBETROTTER v3, EIGENSOFT v5.0.1. Details of parameters used in particular analyses are in the relevant section of Methods or Supplementary Information.

## Data availability

Data for the Spanish cohort were previously published[38] and the individual-level genotype data were used here with permission from those authors. While these data are not yet made openly accessible, they can be made available for academic research on request. All other data sets used in the paper have been previously published, and are publicly available for research use. Specifically, European samples were sourced from POPRES[41] (dbGaP Study Accession: phs000145.v4.p2; approved project number: 6507); north African samples were sourced from a previous publication[30] (accessed via http://www.biologiaevolutiva.org/dcomas/north-african-affy-6-0-data-henn-et-al-submitted/); sub-Saharan African samples from Hapmap Phase 3[31] (accessed via ftp://ftp.ncbi.nlm.nih.gov/hapmap/phase_3/).

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

## Acknowledgements

We acknowledge support from the Wellcome Trust (203141/Z/16/Z, 090532/Z/09/Z, 098387/Z/12/Z, 095552/Z/11/Z, 212284/Z/18/Z) and Fondo de Investigación Sanitaria (Grants PI13/01136 and PI16/01057). We thank G. Hellenthal, D. Lawson and G. Busby for advice on the use of fineSTRUCTURE and GLOBETROTTER. Also G. Hellenthal for providing computer code for the analysis of cluster assignment uncertainty in the fineSTRUCTURE analysis. We also thank M. Robinson, F. Dubert García, and R. Villares Paz for providing background on the history of the Iberian peninsula and advice on historical sources. The support of the Spanish National Gentyping Center (CEGEN-PRB2) is also acknowledged with appreciation.

## Author contributions

A.C., P.D. and S.M. initiated and designed the study. C.F.-R., C.R.-P. and I.Q.-G. collected the data for the Spanish cohort and carried out genotype calling under the direction of A.C. C.B. performed the analyses under the joint supervision of S.M. and P.D. A.C. provided historical information. C.B., A.C, P.D. and S.M. wrote the manuscript.

## Additional information

**Competing interests:** S.M. is a director of GENSCI limited. P.D. is a director and Chief Executive Officer of Genomics plc, and a partner of Peptide Groove LLP. The remaining authors declare no competing interests.

