## [Peer Review File · Nature Communications]

Reviewer #1 (Remarks to the Author):

I reviewed this paper previously, and the authors have satisfactorily addressed my concerns.

Reviewer #2 (Remarks to the Author):

My evaluation from the previous round is largely unchanged. This manuscript still contains interesting analyses about population structure and its possible relationship to history, and the analysis is thorough and competent, but I still have issues with how the results are discussed.

In short, I think that the authors need to do a better job of putting the findings in the context of previous work.

1)

a)

The intro claims:

"We identify extensive fine-scale structure down to unprecedented scales, less than 10 Km in some places."

As I stated in the earlier round, the scale is not unprecedented. Cavalli-Sforza in the 1970s identified structure at even finer scales. I do understand that the present study has more power than studies in the 1970s, and can make more refined claims, e.g., about specific pairs of valleys. However, Cavalli-Sforza could correctly have claimed

"We identify extensive fine-scale structure down to scales of less than 5Km."

Either the authors must clarify what is unprecedented (as they have done in their reply to the earlier round of review) or, if this is too complicated for the intro, simply leave novelty claims for the main text where they can be properly stated.

b)

l. 228: "Our results go beyond this, showing that by leveraging information genome-wide, it is possible to consistently map specific individuals back to their region of origin, at scales <10km, and without utilizing prior geographic information. "

I don't know what is meant by "Consistently" or "at scale". "consistently" is somewhat misleading, because it suggests that this can be done as a matter of routine for individuals. As far as I can tell, all clusters represented have a range of at least 25km. Maybe the authors mean that "Some of the identified clusters have a limited geographical range, with all member individuals falling within a corridor of width below 10km?" Then the "scale <10km" referring to the dimension of least error when mapping individuals to the cluster centroid? Or is it RMS distance from cluster centroid?

c)

As far as I can tell, all these claims are sample-dependent, and inhomogeneous sampling can lead to large underestimates of mapping uncertainty.

2)

l.71:

I also expressed a concern in the previous version that sampling biases could have affected differences in inferred structure across regions. I must not have been clear, since the authors provide a detailed answer that does not address the point I was trying to make. I agree that the authors have shown that population structure exists and I agree that it's not purely a matter of sampling density. The question was whether differences in the amount of population structure across regions can be taken as meaningful, because the sampling strategies (and thus biases) may have been different. This is not a major point in the manuscript, but I think that sampling issues

deserve more discussion.

3)

I.92: "Theoretical arguments predict (Methods) that this effect can only occur if admixture from a highly drifted group into another population takes place, and implies directionality of this admixture"

This is a relatively minor point, but the authors have not addressed my concern from the previous round. Maybe I was not clear.

Even though I understand what the argument should be, I still don't see where this argument is made in the Methods or Supplement. The Methods (I.532-563) barely mentions this. The supplementary section 5 only explains why a scenario with no admixture is implausible and mentions nothing about directionality.

4)

I.130: "For all six Iberian clusters the largest contribution comes from France (63 - 91%), with smaller contributions that relate to present-day Italian (5 - 17%) and Irish (2 - 5%) groups. With the exception of the Basque cluster, these three donor groups dominate"

Assuming that this refers to figure S3b, I don't really see this. The statement appears to actually be true for the Basque, but the North African component seems to dominate the Irish in many other groups.

5)

Figure 3b: I would have expected the best two-way model to include a European and a North Moroccan population. The argument for why the North Moroccan population gets classified as a (non-contributing) major population is fine, but it would help to make it more explicit. Could the authors clarify this? I'm assuming that the answer is something like: "Globetrotter does not allow for two-way migrations, and thus fails to identify this closer relationship". Would globetrotter timing estimates be affected by such misspecification of the source population?

6)

Discussion: I think that in a few cases using political "conquest" rather than demographic "invasion" would have been preferable - I'll leave this up to the authors.

7)

a)

I.195:

"Our analysis, alongside previous smaller studies imply a substantial and regionally varying genetic impact."

I think that the implication was clear enough from the previous studies.

"Our analysis confirms and refines previous findings of a substantial and regionally varying genetic impact"

b)

"Our results further imply that north west African-like DNA predominated in the migration."

I think that this was implied by the historical record, shown by previous DNA studies, and confirmed here.

c)

Line 200:

"Within Spain, north African ancestry occurs in all groups, although levels are low in the Basque region [...] this implies that the Basques have not been completely isolated from the

rest of Spain over the past 1300 years”

The first part was well documented (e.g., in Reference 13). I’m not sure what to do with the second part -- did anyone suggest absolute isolation of the Basques over 1300 years? If so, a citation would be useful.

Minor points/typos

8)

I find the scale on Figure 3b hard to read quantitatively. A bar chart may do a better job?
I could not understand the “4-10%” at the bottom until I read the main text.

Reviewer #1 (Remarks to the Author):

I reviewed this paper previously, and the authors have satisfactorily addressed my concerns.

Reviewer #2 (Remarks to the Author):

My evaluation from the previous round is largely unchanged. This manuscript still contains interesting analyses about population structure and its possible relationship to history, and the analysis is thorough and competent, but I still have issues with how the results are discussed.

In short, I think that the authors need to do a better job of putting the findings in the context of previous work.

1)

a)

The intro claims:

“We identify extensive fine-scale structure down to unprecedented scales, less than 10 Km in some places.”

As I stated in the earlier round, the scale is not unprecedented. Cavalli-Sforza in the 1970s identified structure at even finer scales. I do understand that the present study has more power than studies in the 1970s, and can make more refined claims, e.g., about specific pairs of valleys. However, Cavalli-Sforza could correctly have claimed

“We identify extensive fine-scale structure down to scales of less than 5Km.”

Either the authors must clarify what is unprecedented (as they have done in their reply to the earlier round of review) or, if this is too complicated for the intro, simply leave novelty claims for the main text where they can be properly stated.

We accept the Reviewer’s comment and have removed the word “unprecedented” from the introduction. The relevant sentence now reads:

“We identify extensive fine-scale structure down to scales, less than 10 Km in some places.”

As suggested by the reviewer, we leave further explanation for the discussion.

b)

I. 228: “Our results go beyond this, showing that by leveraging information genome-wide, it is possible to consistently map specific individuals back to their region of origin, at scales <10km, and without utilizing prior geographic information. “

I don't know what is meant by "Consistently" or "at scale". "consistently" is somewhat misleading, because it suggests that this can be done as a matter of routine for individuals. As far as I can tell, all clusters represented have a range of at least 25km. Maybe the authors mean that "Some of the identified clusters have a limited geographical range, with all member individuals falling within a corridor of width below 10km?" Then the "scale <10km" referring to the dimension of least error when mapping individuals to the cluster centroid? Or is it RMS distance from cluster centroid?

We fully accept the Reviewer's concerns about the lack of precision in our previous wording. We have tightened and clarified the wording in our revision, in particular by removing the word 'consistently' and being precise about our meaning of 'range' of a cluster. Specifically, for three of the clusters shown in Figure 2a, the RMS from the cluster centroid is <10 Km. That is, green diamonds; black diamonds; and dark-blue diamonds. We have adjusted the text as follows:

We show that population structure exists at ultra-fine scales in Galicia (Figure 2a), particularly in the province of Pontevedra, with some clusters having geographic ranges of less than 10 Km (root-mean-square from cluster centroid). To our knowledge, these results represent the finest scales over which such structure has yet been observed in humans. Previously, it has been shown that by jointly analysing people from a priori defined sampling locations, subtle differences in group averages at certain genomic loci can be observed at fine geographic scales. For example, subtle differences in blood group frequencies have been observed among villages in Italy's 70km-long Parma Valley²⁵. Our results go beyond this, showing that by leveraging information genome-wide, it is possible to detect subtle genetic structure at fine geographic scales without utilizing prior geographic information.

c)

As far as I can tell, all these claims are sample-dependent, and inhomogeneous sampling can lead to large underestimates of mapping uncertainty.

We agree with the Reviewer's comment and have added a caveat along these lines in the discussion (see below).

2)

I.71:

I also expressed a concern in the previous version that sampling biases could have affected differences in inferred structure across regions. I must not have been clear, since the authors provide a detailed answer that does not address the point I was trying to make. I agree that the authors have shown that population structure exists and I agree that it's not purely a matter of sampling density. The question was whether differences in the amount of population structure across regions can

be taken as meaningful, because the sampling strategies (and thus biases) may have been different. This is not a major point in the manuscript, but I think that sampling issues deserve more discussion.

We had indeed misunderstood the Reviewer's earlier comment and are grateful for the clarification. We accept the concern and have added a paragraph to the main text discussion to make the limitations clearer (below), as well as included a discussion about potential impacts of sampling bias in the Supplementary Information (see Section 2.3 "Discussion of potential sampling effects").

"It is worth noting that differences in the amount of population structure observed in different regions may be sample-dependent (see Supplementary Information for discussion). In principle, if sampling is differentially biased towards, for example, rural versus urban areas in different parts of Spain it could potentially lead to differences in detected patterns of structure. This might mask structure in some regions, but crucially, our approach would not be able to find structure if it was not there."

3)

I.92: "Theoretical arguments predict (Methods) that this effect can only occur if admixture from a highly drifted group into another population takes place, and implies directionality of this admixture"

This is a relatively minor point, but the authors have not addressed my concern from the previous round. Maybe I was not clear.

Even though I understand what the argument should be, I still don't see where this argument is made in the Methods or Supplement. The Methods (l.532-563) barely mentions this. The supplementary section 5 only explains why a scenario with no admixture is implausible and mentions nothing about directionality.

Apologies for the confusion here, and we thank the Reviewer for pointing out gaps in our argument. Some of this may stem from our use (which we now see as confusing) of the word "directionality". We were aiming to make the point that on the basis of the reasoning in the supplementary, that there must have been admixture from the Basque-like cluster into the central cluster. We were not trying to argue that we could also rule out admixture in the *opposite* direction. We have reworded the main text to make this clear (below), and clarified our arguments in Section 5 of the Supplementary Information.

"Theoretical arguments predict (Methods) that this effect can only occur if admixture from a highly drifted group into another population takes places. That is, the effect could not be explained only by Basques inheriting DNA from ancestors of the central group (although this may have happened in addition). Thus, this signal provides

evidence of admixture into the 'central' cluster from a group related to the Basque populations."

4)

I.130: "For all six Iberian clusters the largest contribution comes from France (63 - 91%), with smaller contributions that relate to present-day Italian (5 - 17%) and Irish (2 - 5%) groups. With the exception of the Basque cluster, these three donor groups dominate"

Assuming that this refers to figure S3b, I don't really see this. The statement appears to actually be true for the Basque, but the North African component seems to dominate the Irish in many other groups.

We thank the reviewer for pointing this out. We have adjusted the text to say:

"For all six Iberian clusters the largest contribution comes from France (63 - 91%), with smaller contributions that relate to present-day Italian (5 - 17%) and Irish (2 - 5%) groups. With the exception of the Basque cluster, these three donor groups contribute proportionally similar amounts throughout Iberia,..."

5)

Figure 3b: I would have expected the best two-way model to include a European and a North Moroccan population. The argument for why the North Moroccan population gets classified as a (non-contributing) major population is fine, but it would help to make it more explicit. Could the authors clarify this? I'm assuming that the answer is something like: "Globetrotter does not allow for two-way migrations, and thus fails to identify this closer relationship". Would globetrotter timing estimates be affected by such misspecification of the source population?

That's correct. In any one GLOBETROTTER analysis, only admixture events within the target group (i.e. Iberia) are considered, and a mixture of modern-day 'donor' groups are inferred as proxies for the historical admixing populations. Reciprocal admixture (i.e. when a donor group has some, more recent ancestry *from* the target group) is not explicitly modelled. However, in such a case it just means that the donor group will not be a very good proxy for the historical admixing source and GLOBETROTTER will realise this because the admixture signal (at least from the set of models that GLOBETROTTER uses) will not be strong when that donor group is included. In our context, GLOBETROTTER is unable to infer that only a portion of the north Moroccan genomes are like the source population, and goes for the closest fit instead. However this doesn't affect date estimation for the Iberians because only the receiving of DNA contributes to date estimation. Details of simulations for cases where donors are poor proxies for source populations are in the main GLOBETROTTER paper (reference 20), on pg. 15 and 41 of Supplementary Information, and show that date estimation is reliable in this context.

We have included the following sentence in the results section of the main manuscript, pg 9.

“If this European-like ancestry had arrived more recently than the detected admixture event, the north Moroccan donor group would be a poor proxy for the historical source population and GLOBETROTTER uses a better alternative. Since GLOBETROTTER detects admixture based on the DNA received by the target population (Iberia) this would not affect the date estimates²⁰.”

6)

Discussion: I think that in a few cases using political “conquest” rather than demographic “invasion” would have been preferable – I’ll leave this up to the authors.

We have changed the instances of “invasion” to “conquest” throughout the manuscript.

7)

a)

I.195:

“Our analysis, alongside previous smaller studies imply a substantial and regionally varying genetic impact.”

I think that the implication was clear enough from the previous studies.

“Our analysis confirms and refines previous findings of a substantial and regionally varying genetic impact” .

We have changed the text as suggested by the Reviewer.

b)

“Our results further imply that north west African-like DNA predominated in the migration.”

I think that this was implied by the historical record, shown by previous DNA studies, and confirmed here.

We have adopted the wording suggested by the Reviewer and expanded the text to explain the methodological difference between our approach and previous studies.

Our analysis confirms and refines previous findings^{11,22} of a substantial and regionally varying genetic impact, narrowing to a period spanning < 400 years. Crucially, unlike previous genetic studies of admixture in Iberia^{11,22,25}, we avoid strong assumptions about the genetic make-up of the historical admixing groups. Instead of specifying in advance the modern-day sources that we assume best represent the historical populations that came together in the past, we infer the best mixture of modern-day

populations from a large set of possible groups. Our GLOBETROTTER results suggest that amongst the 6 potential African populations in our study, the best match to the predominant group involved in the actual admixture event is north west African.

c)

Line 200:

“Within Spain, north African ancestry occurs in all groups, although levels are low in the Basque region [...] this implies that the Basques have not been completely isolated from the rest of Spain over the past 1300 years”

The first part was well documented (e.g., in Reference 13). I’m not sure what to do with the second part -- did anyone suggest absolute isolation of the Basques over 1300 years? If so, a citation would be useful.

We accept the Reviewer’s concern and have adjusted the text to say:

“...Therefore, although genetically distinct^{22,23}, north African-like ancestry in the Basque region could be explained through genetic interactions between the Basque groups and other parts of Spain within the past 1300 years.”

Minor points/typos

8)

I find the scale on Figure 3b hard to read quantitatively. A bar chart may do a better job?

I could not understand the “4-10%” at the bottom until I read the main text.

Many thanks for this comment. Adequately capturing all the information visually is complicated. We have amended the figure to show explicitly the estimated proportions of each side of the admixture event, and added a table to the Supplementary Material (Table S4) with all the individual component numerical estimates.

We have also added the following data availability statement to the main manuscript:

Genotype data for the Spanish cohort can be made available for academic research on request. All other data sets used in the paper have been previously published, and are publicly available for research use (see Methods). All figures make use of genotype data for the Spanish cohort, except Figure 3d and Figure 4a.

l.71:

I also expressed a concern in the previous version that sampling biases could have affected differences in inferred structure across regions. I must not have been clear, since the authors provide a detailed answer that does not address the point I was trying to make. I agree that the authors have shown that population structure exists and I agree that it's not purely a matter of sampling density. The question was whether differences in the amount of population structure across regions can be taken as meaningful, because the sampling strategies (and thus biases) may have been different. This is not a major point in the manuscript, but I think that sampling issues deserve more discussion.

3)

l.92: "Theoretical arguments predict (Methods) that this effect can only occur if admixture from a highly drifted group into another population takes place, and implies directionality of this admixture"

This is a relatively minor point, but the authors have not addressed my concern from the previous round. Maybe I was not clear.

Even though I understand what the argument should be, I still don't see where this argument is made in the Methods or Supplement. The Methods (l.532-563) barely mentions this. The supplementary section 5 only explains why a scenario with no admixture is implausible and mentions nothing about directionality.

4)

l.130:" For all six Iberian clusters the largest contribution comes from France (63 - 91%), with smaller contributions that relate to present-day Italian (5 - 17%) and Irish (2 - 5%) groups. With the exception of the Basque cluster, these three donor groups dominate"

Assuming that this refers to figure S3b, I don't really see this. The statement appears to actually be true for the Basque, but the North African component seems to dominate the Irish in many other groups.

5)

Figure 3b: I would have expected the best two-way model to include a European and a North Moroccan population. The argument for why the North Moroccan population gets classified as a (non-contributing) major population is fine, but it would help to make it more explicit. Could the authors clarify this? I'm assuming that the answer is something like: "Globetrotter does not allow for two-way migrations, and thus fails to identify this closer relationship". Would globetrotter timing estimates be affected by such misspecification of the source population?

6)

Discussion: I think that in a few cases using political "conquest" rather than demographic "invasion" would have been preferable - I'll leave this up to the authors.

7)

a)

l.195:

"Our analysis, alongside previous smaller studies imply a substantial and regionally varying genetic impact."

I think that the implication was clear enough from the previous studies.

"Our analysis confirms and refines previous findings of a substantial and regionally varying genetic

impact”

b)

“Our results further imply that north west African-like DNA predominated in the migration.”

I think that this was implied by the historical record, shown by previous DNA studies, and confirmed here.

c)

Line 200:

“Within Spain, north African ancestry occurs in all groups, although levels are low in the Basque region [...] this implies that the Basques have not been completely isolated from the rest of Spain over the past 1300 years”

The first part was well documented (e.g., in Reference 13). I’m not sure what to do with the second part -- did anyone suggest absolute isolation of the Basques over 1300 years? If so, a citation would be useful.

Minor points/typos

8)

I find the scale on Figure 3b hard to read quantitatively. A bar chart may do a better job?
I could not understand the “4-10%” at the bottom until I read the main text.